# A multiscale comprehensive index for evaluating the development level of surface granite discontinuities

Guojun Liu[1,2*], Xiang Wei[3], Zhenkun Hou[4]

1 Engineering Research Center of Development and Application of Ceramsite Concrete Technology, Hunan City University, Yiyang, Hunan, P.R. China, 2 Key Laboratory of Green Building and Intelligent Construction in Higher Educational Institutions of Hunan Province, Hunan City University, Yiyang, Hunan, P. R. China, 3 Research Center for Waste Oil Recovery Technology and Equipment, Chongqing Technology and Business University, Chongqing, China, 4 School of Civil and Transportation Engineering, Guangdong University of Technology, Guangzhou, China

* liuguojun@hncu.edu.cn

## Abstract

The Gansu Beishan area is a preselected candidate site for a high-level radioactive waste repository in China. The development of surface rock mass discontinuities in this region is crucial for ensuring the long-term stability and safety of the project. The survey line method was employed to investigate these discontinuities. Fault geometry information was defined based on the characteristics of traffic routes and exploratory trench wall fault gouge. Optimal joint sets were identified using rose diagram equal-area upper hemisphere projection methods. Statistical analysis shows that the dominant joint orientations in each group follow a normal distribution. Using the circular sampling window theory, the mean trace length and trace midpoint density of joints for each outcrop were calculated. A Multiscale Discontinuity Comprehensive (MDC) index was proposed to evaluate the degree of surface rock mass discontinuity development based on discontinuity geometry parameters. The results of the surface rock mass discontinuity development were analyzed according to the tectonic stress and mechanical formation mechanisms of the discontinuities. These research findings provide critical data to support the ongoing development of high-level radioactive waste geological disposal.

## 1. Introduction

In the geological disposal of high-level radioactive waste (HLW), nuclide migration is a critical factor in assessing the safety and reliability of the repository. The structural surface serves as the primary pathway for nuclide diffusion into the biosphere through surrounding rock and groundwater [1]. Therefore, understanding the development characteristics of the structural surface is essential for site selection and the evaluation of HLW repositories.

**Data availability statement:** All relevant data are within the manuscript. The figures and tables used to support the findings of this study are included in the article.

**Funding:** Chongqing Education Commission Science and Technology Research project (No. KJQN202400837), Aid Program for the Science and Technology Innovative Research Team in Higher Educational Institutions of Hunan Province.

**Competing interests:** The authors declare that they have no conflicts of interest.

The development characteristics of structural surfaces are primarily defined by their geometric features. Currently, the geometric parameters of structural surfaces are mainly obtained through field investigations and quantitatively described using probabilistic statistical methods. In 1978, the International Society of Rock Mechanics (ISRM) specified 10 descriptive indices for rock mass structural surfaces, which were updated in 2007 [2]. Schmidt [3] introduced the use of Schmidt diagrams to describe the regularity of nodal orientations and classify nodal dominance groups. Since then, both domestic and international scholars have widely adopted Schmidt diagrams as a fundamental tool for classifying nodal dominance groups, often supplementing them with their own methods and analyses. For instance, Shanley R.J. [4] proposed using cluster analysis to classify nodal dominance groups, implementing this through computational techniques. Tian Kaiming [5] explored the classification of nodal dominance groups, applying Fisher distribution as an objective function for testing. Prist and J.A. Hudson [6,7] were pioneers in applying the line measurement method to analyze the relationship between line measurements and nodal distribution on outcrops, as well as suggesting theoretical distributions for trace length and methods for estimating the average trace length. L. Zhang and H.H. Einstein [8], along with M. Mauldon [9], focused on calculating point surface density in traces using the circular window method. P.H.S.W. Kulatilake and T.H. Wu [10] examined the possibility of the midpoint of a trace falling within the window under an assumed distribution of trace lengths, though the actual distribution remains uncertain. Chun Yang [11] applied principles of the circular sampling window method to calculate both the average trace length and trace midpoint surface density. More recently, Ge, Y., Cao, B., and Tang, H. [12] proposed the use of Artificial Neural Networks to identify discontinuities from 3D point clouds. Farmakis I. [13] utilized LiDAR terrestrial laser scanning for rockfall susceptibility analysis, evaluating and characterizing 3D jointed rock mass structures.

Research on the developmental characteristics of structural facets has progressed based on the geometric properties identified by both domestic and international scholars. In 1964, Deere [14] introduced the Rock Quality Designation (RQD) index, which became a fundamental parameter for assessing rock mass integrity. Later, in 1973, Bieniawski [15] developed the Rock Mass Rating (RMR) system for evaluating rock engineering quality, incorporating RQD and joint spacing as key indicators of rock joint development. Building on this, Barton [16] introduced the Q-system, a quantitative classification method for rock engineering quality, where the ratio RQD/J is used to characterize joint development.

While these studies primarily focus on structural facets (joints) at the statistical scale, larger-scale structural facets (faults) at regional and engineering scales play a crucial role in controlling the stability and continuity of engineering rock masses. Therefore, a multiscale structural facet synthesis approach is essential for accurately evaluating the degree of structural facet development in rock masses.

China's geological disposal project for high-level radioactive waste follows a three-step implementation process: repository site selection and evaluation, underground test chamber construction, and final repository construction. During the site selection and evaluation phase, understanding the development of structural surfaces in the

preselected area is critical. This paper presents a detailed site investigation of granite rock formations surrounding boreholes BS22 and BS23 in the preselected Bei Shan area of Gansu Province. The study involved analyzing nodal dominance groups, average trace length, and trace point surface density in outcrops around each borehole to infer the location and extent of fault surfaces in the vicinity. Based on the determined structural surface geometry parameters, the Multiscale Discontinuity Comprehensive (MDC) index method was applied to assess the degree of structural surface development in the surface rock around the two boreholes. Additionally, the study examined the mechanical genesis of these structures, providing essential data for the site selection of the high-level radioactive waste repository.

## 2. Structural surface survey methods

The Calculated Well sub-lot (Fig 1) is one of the favorable candidate sites in the preselected Bei Shan area of Gansu Province for the geological disposal of high-level radioactive waste in China. The site is characterized by a dry, water-scarce environment with no perennial flowing water, a typical continental climate, arid and windy conditions, and sparse, undeveloped vegetation. The region is sparsely populated, with most residents being nomadic Mongolian herdsmen. These factors contribute to favorable geological and hydrological conditions for a high-level radioactive waste disposal site. However, the development of structural surfaces within the surface rock of the Calculated Well sub-lot presents a challenge for repository construction (Fig 2). Therefore, it is essential to obtain spatial distribution parameters of the surface rock structure through field geological investigations, assess the degree of structural surface development, and provide critical data for site comparisons in the repository selection process.

The surface lithology of the rock mass in the Beishan pre selection area of Gansu Province is single, mainly granite, with extensive development of bedrock in large areas, ranging in thickness from several kilometers to tens of kilometers, but it can basically meet the depth requirements for high-level radioactive waste geological disposal. Granite has the following characteristics: 1) low porosity (about 0.5%), low permeability coefficient ($10^{-7} \sim 10^{-9}$ m·s$^{-1}$), low moisture content

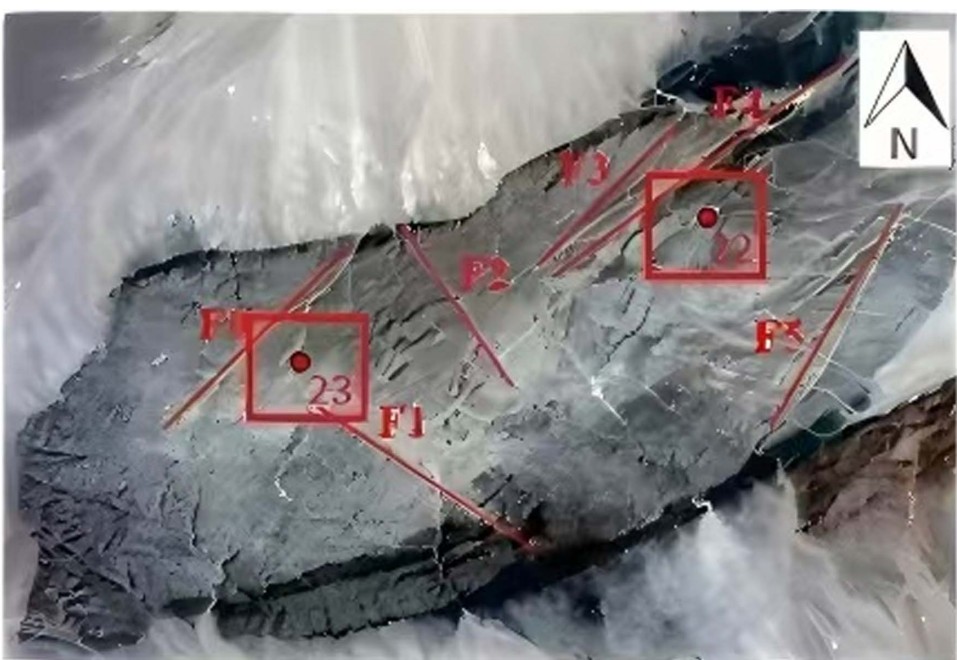

**Fig 1. Joint investigation and faults distribution for Suanjingzi section [ 17]. (A)** F0, F1, F2, F3, F4, F5 respectively represent faults 1-5. **(B)** The numbers 22 and 23 represent the borehole numbers. (C) The scale is 1:500.

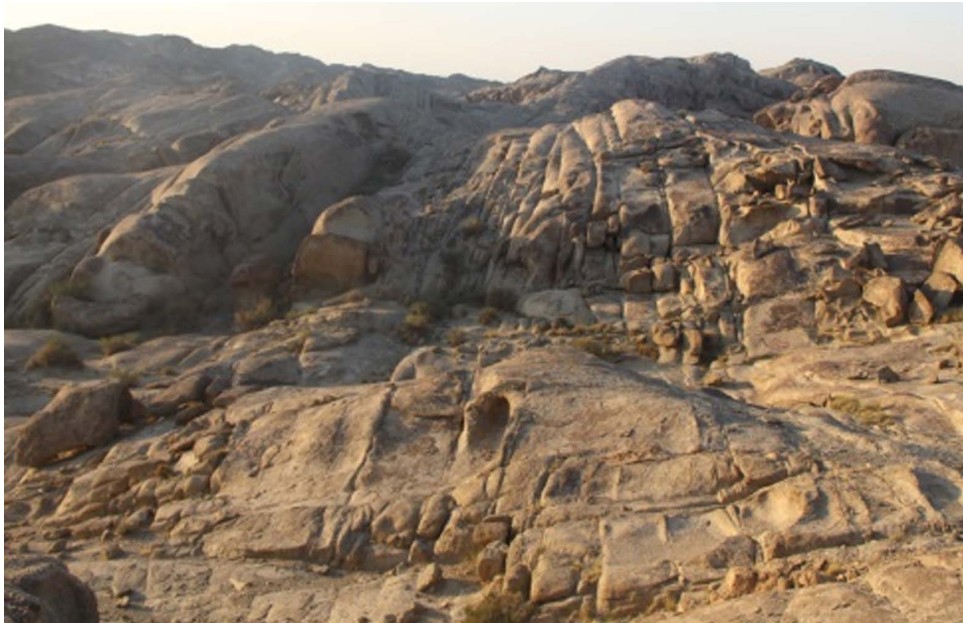

**Fig 2. Joint development in surface rock mass [ 17].** This picture is a field photo taken of the joint development in the Suanjingzi section.

(0.1%~0.2%), and high strength (uniaxial compressive strength above 150MPa); 2) The mechanical strength is high, and the geological disposal of high-level radioactive waste has a relatively small impact on the surrounding rock; 3) Good thermal conductivity, with minimal impact from the decay heat of spent fuel; 4) Good radiation resistance, and the physical and mechanical properties of rocks remain unchanged after radiation exposure; 5) The most crucial factor is that granite has high resistance to nuclides. Given these characteristics of granite, it will be the preferred surrounding rock for China's high-level radioactive waste geological disposal facility.

### 2.1. Nodal survey method

The scope of the nodal survey covered detailed and accurate field measurements of the outcrops within the 4 km² area surrounding drill holes BS22 and BS23 (Fig 1). The survey was conducted using the line survey method (Fig 3). Joints were classified into three types based on their relationship with the survey line: Type I (intersecting), Type II (extension line intersecting), and Type III (non-intersecting). The geometric parameters of each joint within the outcrop were measured. To ensure both accuracy and efficiency, the survey lines were strategically arranged to intersect all nodal traces as much as possible.

### 2.2. Fault investigation methods

Based on the characteristics of the fault mud observed on the wall of the exploratory trench (Fig 4), the formation of the fault was inferred. Its spatial extension was subsequently traced and investigated using remote sensing maps and GPS, allowing for an assessment of the scale of its development.

### 2.3. Statistics and analysis of structural surface geometric features

Joints are commonly found in rock masses, typically exhibiting a random distribution. They not only influence the fundamental mechanical properties of these masses but also play a significant role in engineering stability. Joints are products

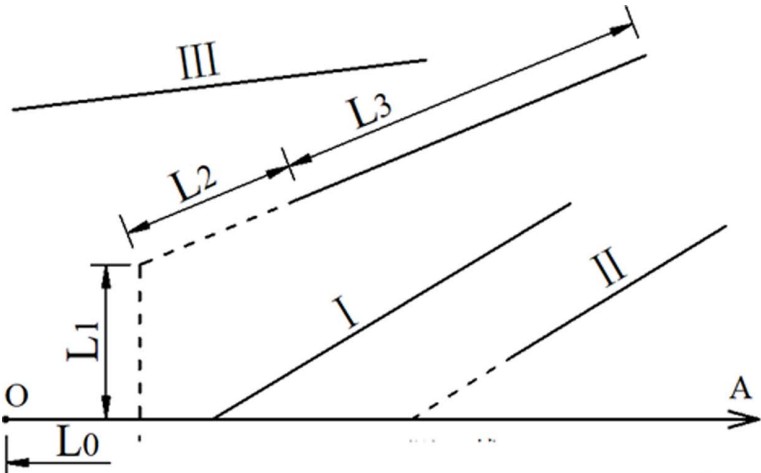

**Fig 3. Joint investigation through Survey line method. (A)** L0-L3 represents the length of joints 0-3. **(B)** OA solid line represents the survey line. **(C)** I (intersecting type), II (extension line intersecting type), III (non-intersecting type).

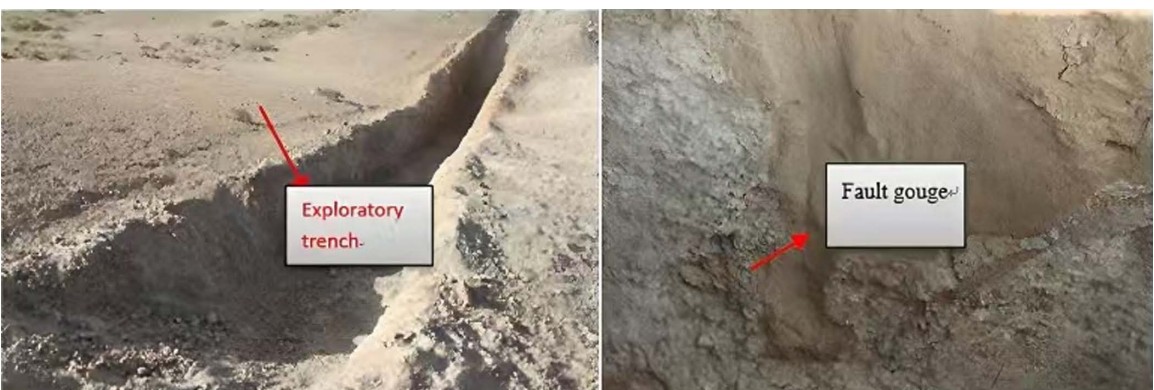

**Fig 4. Exploratory trench and fault gouge [ 17].** Process diagram for investigating fault occurrence.

of rock formations shaped by specific geological environments and tectonic stress fields, and their spatial distribution follows statistical patterns under certain stress conditions. The geometric characteristics of the joints, especially their formation shape, determine their specific spatial arrangement. In practice, while the occurrence of joints may vary, it generally adheres to certain statistical rules.

Nodal production statistics are primarily based on nodal production illustrations, such as production rose diagrams, equal angle or equal area scatter diagrams, and equal density diagrams, to define dominant groups. From the nodal tendency rose diagram, pole distribution diagram, and nodal pole equal density diagram of borehole BS22 (shown in Fig 5), it is concluded that the surface rock around borehole BS22 develops four groups of nodules at orientations of 35.5°∠74.9°, 119.7°∠73.8°, 215.5°∠71.9°, and 302.4°∠70.1°. The same method was applied to delineate the dominant group of surface rock joints around borehole BS23. Table 1 presents the characteristics of the dominant joint groups in boreholes BS22 and BS23.

Based on the results from the nodal dominant group divisions of each drill hole, the statistical frequency distribution of each dominant group's production was segmented into intervals. Corresponding histograms were then plotted to derive fitting

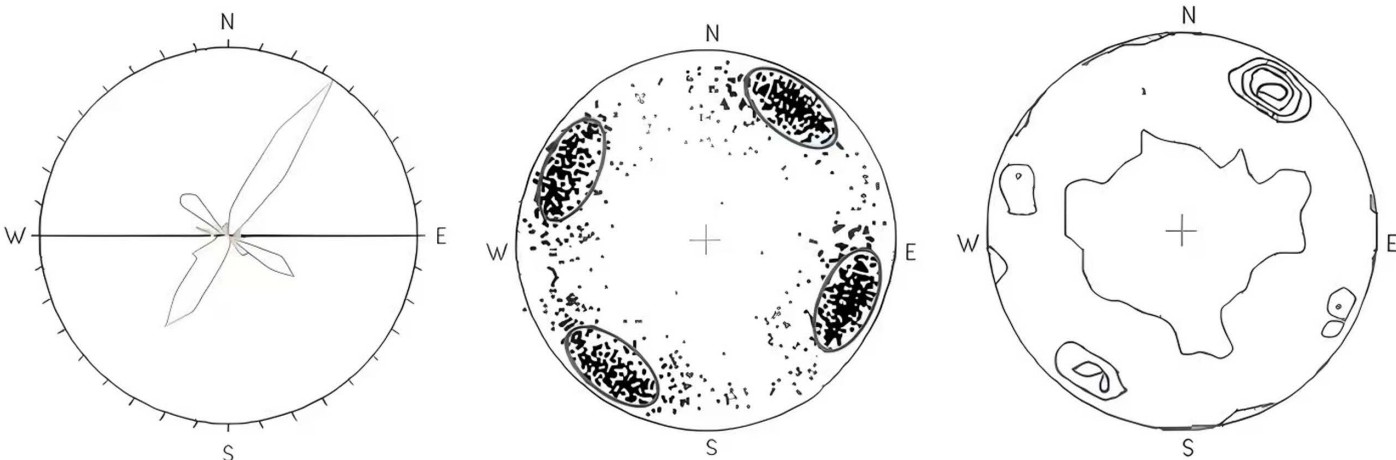

**Fig 5. Diagram of join for BS22 borehole.** The three figures represent from left to right respectively tendency rose diagram, pole distribution diagram, and nodal pole equal density diagram of borehole BS22.

**Table 1. Optimal joint set for BS22and BS23 borehole.**

| Hole number | Advantage Group | Advantageous properties | |
| --- | --- | --- | --- |
| | | Tendency (°) | Inclination (°) |
| BS22 | 1 | 35.5 | 74.9 |
| | 2 | 119.7 | 73.8 |
| | 3 | 215.5 | 71.9 |
| | 4 | 302.4 | 70.1 |
| BS23 | 1 | 48.7 | 75.0 |
| | 2 | 124.6 | 68.4 |
| | 3 | 180.0 | 79.1 |
| | 4 | 249.6 | 76.4 |
| | 5 | 307.8 | 72.7 |

curves and probability density distribution functions. Fig 6 displays the statistical histograms, fitting curves, and fitting equations for the tendency and dip angle of the dominant group in drill hole BS23 at 180.0°∠79.1° in the Counting Well Sub.

From Fig 6, it is evident that the probability distribution of the dominant group tendency and dip angle for borehole BS23 at 180.0°∠79.1° can be effectively modeled using a normal distribution function, represented by the following expressions.

$$y = \frac{1}{\delta\sqrt{\pi/2}} e^{-2\frac{(x-\mu)^2}{\delta^2}}$$

(1)

The same approach was used to statistically analyze the dominant groups in boreholes BS22 and BS23 of the well sub. This led to normal distribution functions that accurately fit the distributions of nodal tendency and dip angle. The parameters for the normal distributions of both the tendency and dip angle for each dominant group are provided in Table 2.

## 2.4. Average trace length of nodes and density of points in traces

Many researchers have explored the use of the circular window method to estimate the density and average trace length of point surfaces in rock nodal traces [18–20]. However, directly applying circular windows to investigate joint geometry

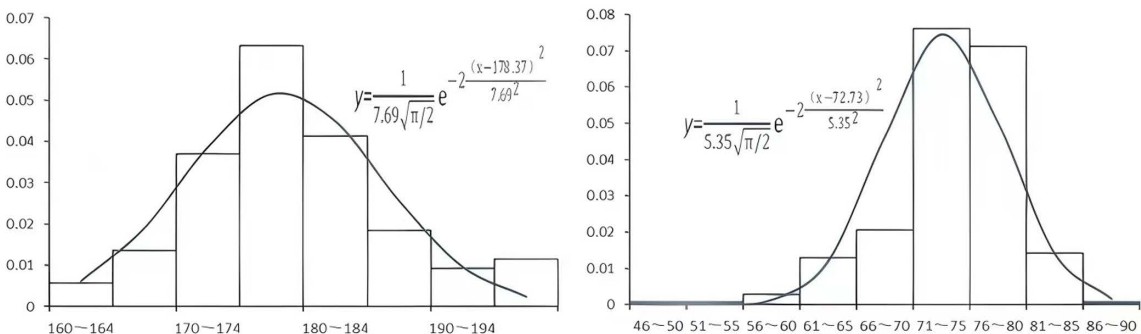

**Fig 6. Histogram, fitting curve and formula of occurrence for optimal set 180.0°∠79.1°. (A)** According to the division results of each drilling joint advantage group, the frequency distribution of the occurrence of each advantage group is statistically analyzed between zones, and its histogram is drawn to obtain the corresponding fitting curve and probability density distribution function. **(B)** Using the same method to statistically calculate the dominant groups of boreholes BS22 and BS23, a normal distribution function can be obtained to effectively fit the distribution of joint inclination and dip angle.

**Table 2. Fitting parameters of optimal joint set for BS22 and BS23 borehole.**

| Hole number | Advantage Group | Propensity to fit parameters | | Inclination fitting parameters | |
|---|---|---|---|---|---|
| | | $\mu$ (°) | $\delta\,\theta$ (°) | $\mu$ (°) | $\delta$ (°) |
| BS22 | 1 | 32.09 | 24.08 | 69.64 | 8.34 |
| | 2 | 117.26 | 27.73 | 69.71 | 7.31 |
| | 3 | 218.33 | 27.45 | 68.21 | 7.91 |
| | 4 | 296.59 | 22.30 | 68.68 | 9.08 |
| BS23 | 1 | 48.29 | 20.14 | 73.07 | 7.96 |
| | 2 | 122.52 | 19.15 | 65.58 | 11.59 |
| | 3 | 178.37 | 7.70 | 72.73 | 5.35 |
| | 4 | 246.18 | 23.27 | 69.71 | 6.63 |
| | 5 | 312.60 | 22.24 | 69.58 | 9.24 |

during nodal surveys is impractical in the field. Instead, nodal measurement data collected via the lineament method are formatted into a standardized data structure {L0, L1, L2, L3}. Using the principles of the circular window method, a MATLAB program was developed to generate the nodal trace distribution map for the sampled outcrop. The outcrop nodal data were then analyzed with circular sampling windows of different radii and locations to calculate the average trace length and midpoint surface density for each outcrop. The specific calculation formulas are presented in equations (2) and (3).

$$v = \frac{\pi\,(N+N_2-N_0)}{2\,(N-N_2-N_0)}\,c \tag{2}$$

$$\lambda = \frac{(N-N_2-N_0)}{2\pi c^2} \tag{3}$$

where: $u$ is the average value of nodal trace length; $c$ is the radius of circular sampling window, $\lambda$ is the density of nodal surface; set the expected number of penetrating traces as $N_0$, the expected number of intersecting traces as $N_1$, the expected number of inclusive traces as $N_2$; the total number of nodal traces as $N_0 = N + N_1 + N_2$.

Outcrops surrounding boreholes BS22 and BS23 with more than 25 joints were selected to ensure a relatively uniform distribution and to prevent skewing the midpoint of the traces, thus improving the accuracy of the calculation results. Using

the circular window method, the midpoint surface density and average trace length of the joints around boreholes BS22 and BS23 were computed. The distribution of point surface density and average trace length for the nodal traces around borehole BS22 is shown in <u>Fig 7</u>.

## 3. Results analysis and engineering applications

### 3.1. Fault geometry statistics

Through a field geological survey and analysis of a remote sensing map (<u>Fig 1</u>), fault $F_1$ was determined to be distributed in a north-west to south-east direction, extending approximately 5.0 km with a fracture orientation of 125°. At the north-west end of fault F1, Trench 1 is developed with fractured granite, a fracture zone, and fault mud. The fault mud is oriented at 26°∠85°, the fracture zone is about 13 m wide, and the fault mud measures approximately 15 cm in width. By using the remote sensing map and GPS tracking to investigate the fault's extension, the geographic coordinates (latitude and longitude) of the starting and ending points of the fault were converted to plane coordinates ($x_1$, $y_1$) and ($x_2$, $y_2$), and the spatial extension length *L of the fault* is shown in formulas (4).

$$L = \sqrt{(x_1 - x_2)^2 + (y_1 - y_2)^2}$$

(4)

where: *L* is the length of the fault; (x1, y1) and (x2, y2) are the plane coordinates of the starting and ending points of the fault, respectively. The comprehensive judgment of fault production is 35°∠85°, and the extension length is 4.49 km.

### 3.2. Evaluation the degree development of the structural surface

In the geological disposal of high-level radioactive waste, two critical concerns arise: the migration and diffusion of radionuclides from the waste into the biosphere, and the stability of the surrounding rock in underground disposal sites. In China, granite is the preferred rock type for high-level waste disposal. Physical and mechanical tests show that granite has low porosity, minimal water content, and poor water absorption, with an average compressive strength of around 170 MPa—well above the 60 MPa threshold for hard rock. As such, the stability of the surrounding rock is generally considered

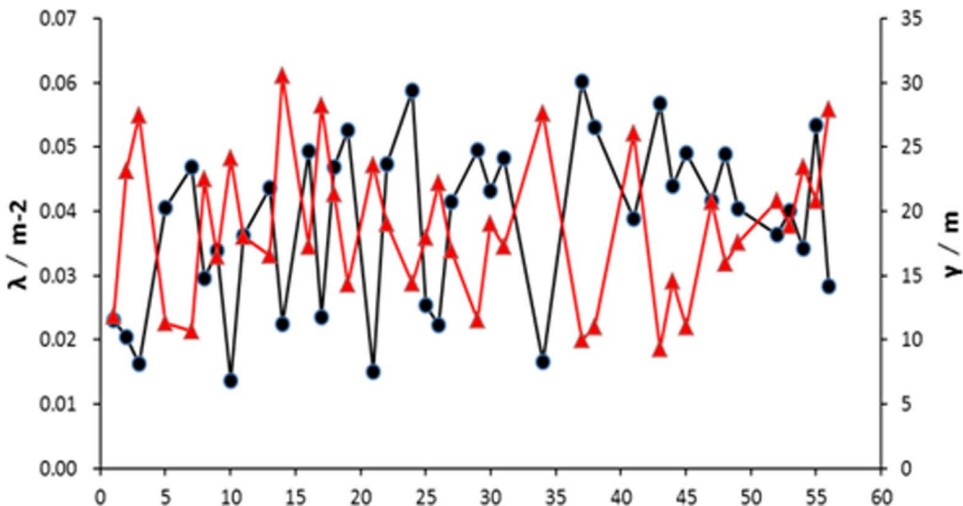

**Fig 7. Distribution of trace midpoint density and mean trace length for BS22 borehole. (A)** Using the circular window method to calculate. **(B)** The red line represents point surface density, the black line represents average trace length for the nodal traces. **(C)** υ is the average value of nodal trace length; λ is the density of nodal surface.

secure. However, granite naturally contains structural surfaces, which act as primary pathways for the migration and diffusion of radionuclides to the biosphere through groundwater. These structural surfaces also weaken the mechanical properties of the rock mass. Therefore, understanding the development characteristics of structural facets in granite is essential for the safe geological disposal of high-level radioactive waste.

### 3.3. Evaluation index of the development degree of structural surface

Structural facets are commonly developed fracture structures in the shallow crust that play a significant role in determining the stability and construction of engineering rock masses. At present, the rock quality designation (RQD) is the primary system used to assess the extent of structural surface development [21,22], while some scholars also consider structural surface spacing or bulk density (Jv). However, these research findings pertain to subsurface rock and primarily focus on the geometric characteristics of joints.

In the high-level radioactive waste disposal project, identifying the characteristics of structural surfaces in surface rock within candidate areas and evaluating their development are critical factors in repository site selection. Therefore, it is crucial to develop a method for assessing the degree of structural surface development in preselected areas for the geological disposal of high-level waste in China.

Considering these factors, and taking into account the evaluation stages of the preselected area for the HLW repository (site selection and site characterization), along with the measured data and the characteristics obtained through data processing, a multiscale structural surface comprehensive index (MDC) is proposed to assess the degree of structural surface development in the surface rock mass. The formulae for this index are as follows.

$$MDC = S + JSR = LW + W_n\lambda\nu \tag{5}$$

where: $S$ is the area of the fault zone; JSR [23] is the evaluation index for the degree of joint development; L is the length of the fault; W is the width of the fault zone; $W_n$ is the number of joint groups; $\lambda$ is is the density of points and surfaces in the trace; and $\nu$ is the length of the joint traces. It is important to note that the parameters on the right side of equation (5) represent their graded scoring values rather than their physical measurements.

Since faults play a decisive role in the continuity of the rock mass, its mechanical characteristics, and groundwater storage, and are also a key factor in the siting and construction of high-level waste disposal projects, a weight of 70% is assigned to the fault (S). Joints primarily influence the connectivity and linkage of the rock mass, so the Joint Surface Ratio (JSR) is assigned a weight of 30%. Therefore, the calculation formula for the MDC becomes [24].

$$MDC = 0.7S + 0.3JSR \tag{6}$$

$$MDC = 0.7\sum_{i=1}^{n} s_i/n + 0.3\sum_{j=1}^{m} JSR_j/m \tag{7}$$

Based on the recommendations of the ISRM [25] and the results of the field survey, Table 3 presents the scoring values for each index, while Table 4 provides the MDC scoring values along with corresponding descriptions of the degree of structural surface development [26,27].

### 3.4. Engineering applications

The geometric parameters of surface rock joints were studied by selecting outcrops with more than 20 joints around boreholes BS22 and BS23 in the Calcineur sub-lot. This selection was made to avoid errors due to insufficient data. The geometric parameters of faults F0 and F1 around borehole BS22, as well as faults F3 and F4 around borehole BS23, were

**Table 3. Rating value of each indicator.**

| $L$(km) | <1 | 1~5 | 5~10 | 10~20 | >20 |
|---|---|---|---|---|---|
| Rating Value | 0~4 | 4~8 | 8~12 | 12~16 | 16~20 |
| $W$(m) | <1 | 1~5 | 5~10 | 10~20 | >20 |
| Rating Value | 0~1 | 1~2 | 2~3 | 3~4 | 4~5 |
| $W_n$(Group) | 0~1 | 2 | 3 | 4 | ≥5 |
| Rating Value | 0~0.8 | 0.8~1.6 | 1.6~2.4 | 2.4~3.2 | 3.2~4.0 |
| $\lambda$ (m$^{-2}$) | 0~0.2 | 0.2~0.6 | 0.6~2 | 2~6 | >6 |
| Rating Value | 0~1 | 1~2 | 2~3 | 3~4 | 4~5 |
| $v$ (m) | 0~1 | 2~3 | 3~10 | 10~20 | >20 |
| Rating Value | 0~1 | 1~2 | 2~3 | 3~4 | 4~5 |

**Table 4. Description of *MDC* score values and the corresponding structural surface development degree.**

| *MDC* | 0~20 | 20~40 | 40~60 | 60~80 | 80~100 |
|---|---|---|---|---|---|
| Grading | I | II | III | IV | V |
| Description of the degree of structural surface development | Low | relatively low | Moderate | high | High |

determined. The development scale of faults F3 and F4 around borehole BS23 was also assessed. The identified structural facies geometric parameters were assigned scores according to the provided table and substituted into equations (5) and (7) to calculate the MDC scoring values. This allowed for an evaluation of the degree of structural facies development in the surface rock mass surrounding the boreholes (Table 5).

From the Table 5, it can be concluded that the degree of structural surface development around borehole BS22 in the Counting Well section is lower than that around borehole BS23. Therefore, it can be tentatively concluded that the surface rock around borehole BS22 is more suitable as a pre-selected site for the construction of the HLW repository.

## 4. Discussion and geomechanical genesis

A structural surface is a fracture structure formed under tectonic stress that disrupts the continuity and integrity of the rock, influencing its mechanical and seepage characteristics. A fracture structure with little or no displacement of rocks along the fracture surface is termed a joint, while a fracture structure with significant displacement along the fracture surface is called a fault. Both faults and joints are commonly developed geological features found in the shallow rocks of the Earth's crust.

Joints form under specific stress conditions. When joints develop under shear stress, they are referred to as shear joints, while those formed under tensile stress are known as tensile joints. Fig 8 illustrates the stress state associated with joint formation. Faults, on the other hand, develop under a triaxial stress state, where two principal stress axes are horizontal and one principal stress axis is perpendicular, or nearly perpendicular, to the horizontal plane. Anderson provides a depiction of the stress state for fault formation, as shown in Fig 9.

Based on the mechanical properties and geometric characteristics of the faults, it is observed that faults F3 and F4 are present around borehole BS22, indicating that the surface rock is subjected to northeast-east (NEE) directional tectonic

**Table 5. *MDC* score values.**

| Hole number | *MDC* | Grade | Description of the degree of structural surface development |
|---|---|---|---|
| BS22 | 16.05 | I | Low degree of structural surface development |
| BS23 | 23.10 | II | Low level of structural surface development |

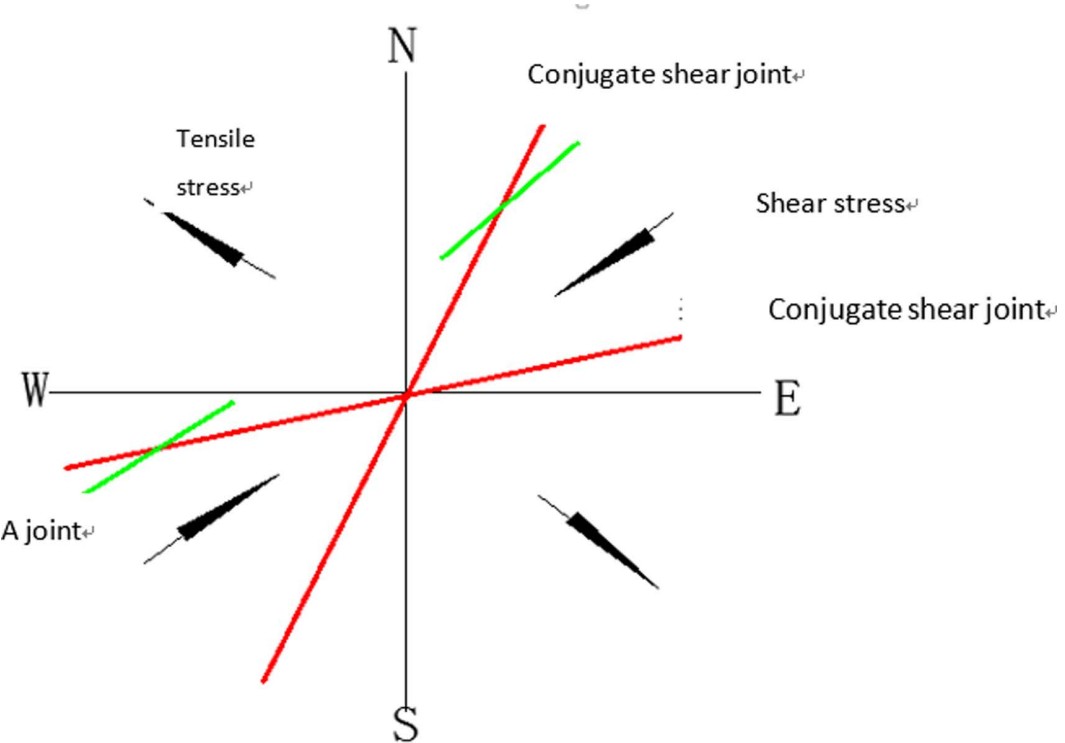

**Fig 8. Stress state of joints.** It shows the stress state of joint formation, According to the mechanical causes of joint formation, joints formed under shear stress are called shear joints, while joints formed under tensile stress are called tensile joints.

stress. Around borehole BS23, faults F0 and F1 suggest the presence of both NEE directional and northwest-west (NWW) tectonic stresses. This aligns with the findings from the regional tectonic stress field, which indicates that the rock mass in the Jingzi section area is influenced by NEE directional tectonic stress. Additionally, the rock around borehole BS23 is subjected to both regional and local stresses.

Joints form under the influence of tectonic stress. The stress state of the surface rock mass around the boreholes suggests that NE-oriented and west-east (W-E) oriented conjugate shear joints are likely to form around borehole BS22, while both NE-oriented and W-E oriented conjugate shear joints, along with NE-oriented tensile joints, are expected around borehole BS23. This explains why a greater number of surface rock joints have developed around borehole BS23 compared to borehole BS22.

Based on the characteristics of the tensor and shear joints, it is evident that the surface rock around borehole BS23 exhibits a dense distribution of joints and shorter joint trace lengths due to the presence of tensile joints. As a result, the density of points in the traces is higher, and the average trace length is shorter compared to the area around borehole BS22. These observations align with the statistical results from the surface investigation.

Faults F0, F3, and F4 are compressional shear fractures formed under regional tectonic stress, while fault F1 is a tensional shear fracture resulting from the combined effects of regional and local stress. As a result, fault $F_1$ has a poor extension and a wide fault zone. Therefore, it can be concluded that the surface rock around borehole BS23 is more developed in terms of joints compared to that around borehole BS22.

## 5. Conclusion

"Based on the detailed geological background of the surface rock masses in the Beijianzi section of Gansu, a comprehensive investigation and statistical analysis of the joint geometric characteristics in the section were conducted through

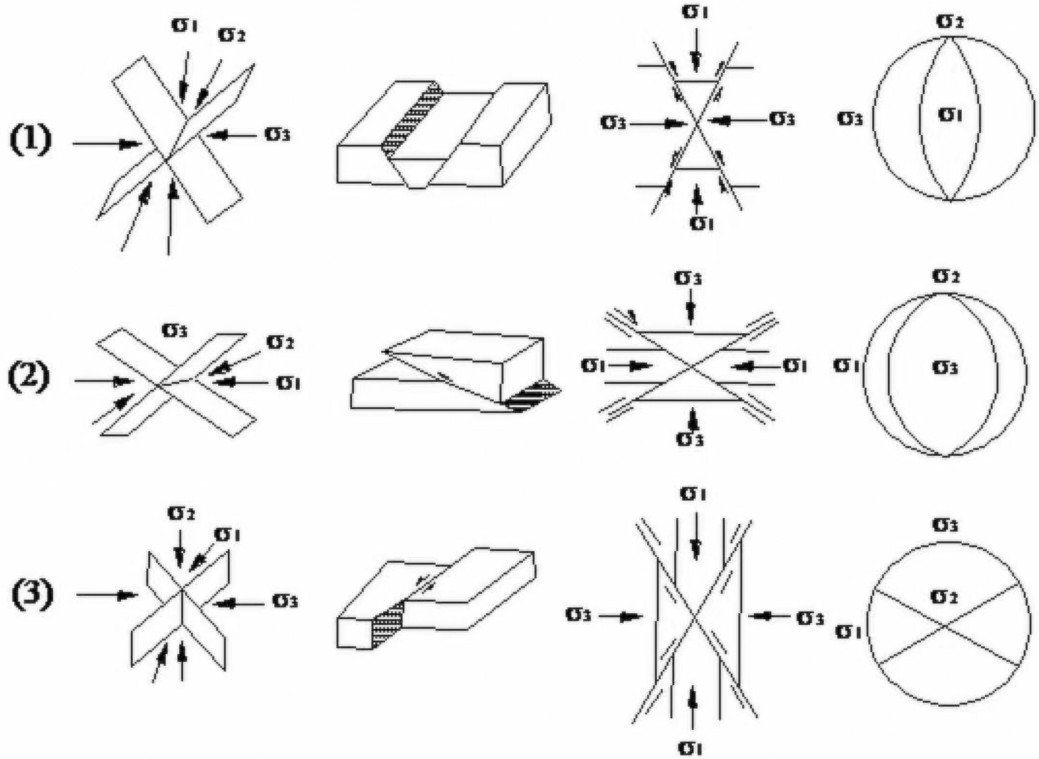

**Fig 9. Stress state of faults.** A fault is formed under triaxial stress conditions, $\sigma_1$ is perpendicular or approximately perpendicular to the horizontal plane, $\sigma_2$ and $\sigma_3$ are horizontal.

field geological surveys using an integrated measurement method. Additionally, information about the geometric properties of the faults was gathered from remote sensing maps and the characteristics of fault mud observed on the walls of the exploratory trench. Building on this data, a multi-scale structural surface composite index method (MDC index) was proposed to assess the degree of structural surface development of the surface rock around boreholes BS22 and BS23, as well as to analyze their mechanical genesis. The main conclusions of the study are as follows:

(1) Nodal Analysis: Utilizing nodal rose diagrams and nodal pole diagrams, it was determined that one nodal dominant group is developed around borehole BS22 and one around borehole BS23 in the Beijianzi section, respectively. The nodal tendency and dip angles conform to a normal distribution. Employing the circular window method, the density and average trace length of nodal traces in each outcrop around boreholes BS22 and BS23 were calculated. Furthermore, through field geological surveys and remote sensing maps, it was established that fault F1 has a dip of 35°∠85° and an extension length of 4.49 km.

(2) MDC Index Evaluation: The evaluation of surface rock development around boreholes BS22 and BS23 using the MDC index indicates that the development around borehole BS22 is less pronounced compared to that around borehole BS23. This suggests that BS22 is more suitable as a pre-selected site for the construction of the high-level waste (HLW) repository.

(3) Mechanical Genesis Analysis: Based on the mechanical genesis of the structural facets, an analysis of the varying geometric parameters of the joints and faults around boreholes BS22 and BS23 was conducted. This analysis highlighted the differences in the degree of structural facet development within the surface rock mass.

The structural surface is critical in the high-level waste repository project. Evaluating the degree of structural surface development in the surface rock mass provides a foundation for further research into groundwater seepage characteristics and nuclide migration patterns. It also contributes to a comprehensive understanding of the integrity and stability of the regional rock mass, offering essential data support for the geological disposal of high-level waste.

## Acknowledgments

The authors would like to show sincere thanks to those techniques who have contributed to this research.

## Author contributions

**Data curation:** Guojun Liu.

**Formal analysis:** Xiang Wei.

**Supervision:** Zhenkun Hou.

**Writing – original draft:** Guojun Liu.

**Writing – review & editing:** Xiang Wei, Zhenkun Hou.

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
