## [Decision Letter · Decision Letter 0]

24 Jan 2025

PONE-D-24-40516A multiscale comprehensive index for evaluating the development level of surface granite discontinuitiesPLOS ONE

Dear Dr. Liu,

Thank you for submitting your manuscript to PLOS ONE. After careful consideration, we feel that it has merit but does not fully meet PLOS ONE’s publication criteria as it currently stands. Therefore, we invite you to submit a revised version of the manuscript that addresses the points raised during the review process.

We look forward to receiving your revised manuscript.

Kind regards,

Fabio Trippetta, Ph.D.

Academic Editor

PLOS ONE

**Journal Requirements:**

the Program for Natural Science Foundation of Hunan Province (No. 2021JJ40028).

6. We note that Figures 1, 2, and 4 in your submission contain copyrighted images. All PLOS content is published under the Creative Commons Attribution License (CC BY 4.0), which means that the manuscript, images, and Supporting Information files will be freely available online, and any third party is permitted to access, download, copy, distribute, and use these materials in any way, even commercially, with proper attribution. For more information, see our copyright guidelines: http://journals.plos.org/plosone/s/licenses-and-copyright.

We require you to either present written permission from the copyright holder to publish these figures specifically under the CC BY 4.0 license, or remove the figures from your submission:

a. You may seek permission from the original copyright holder of Figures 1, 2, and 4 to publish the content specifically under the CC BY 4.0 license. 

Reviewers' comments:

Reviewer's Responses to Questions

**Comments to the Author**

1. Is the manuscript technically sound, and do the data support the conclusions?

Reviewer #1: Yes

Reviewer #2: Partly

2. Has the statistical analysis been performed appropriately and rigorously? 

Reviewer #1: Yes

Reviewer #2: No

3. Have the authors made all data underlying the findings in their manuscript fully available?

Reviewer #1: Yes

Reviewer #2: Yes

4. Is the manuscript presented in an intelligible fashion and written in standard English?

Reviewer #1: Yes

Reviewer #2: Yes

5. Review Comments to the Author

**Reviewer #1: ** The manuscript presents a novel Multiscale Discontinuity Comprehensive (MDC) index for assessing the development level of surface granite discontinuities, which is crucial for the geological disposal of high-level radioactive waste. The study's focus on the Gansu Beishan area, a preselected candidate site for such a repository in China, adds regional significance. The topic is interesting, and this manuscript has a certain degree of innovation; however, some revisions are needed before it can be considered for publication. The detailed comments and suggestions are given as follows:

(1) Why is there a need to propose such a new metric? The explanation is not sufficiently clear.

(2) Figure 9 is missing a legend.

(3) Why is Eq. (5) used to express the new index in this form? Please further explain.

(4) To be frank, the details and depth of analysis in the paper are insufficient. It is recommended to strengthen these aspects further.

(5) “Based on the characteristics of the fault mud observed on the wall of the exploratory trench (Figure 4), the fault's formation was deduced, and its spatial extension was traced and investigated using remote sensing maps and GPS to determine its development scale.” →“Based on the characteristics of the fault mud observed on the wall of the exploratory trench (Figure 4), the formation of the fault was deduced, and its spatial extension was traced and investigated using remote sensing maps and GPS to determine the scale of its development.”

(6) “FIGURE10 Stress state of joint” → “FIGURE 10: Stress State of Joints”

(7) Several key related papers should be considered for inclusion: (1) http://dx.doi.org/10.1016/j.enggeo.2016.03.007; and (2) https://doi.org/10.1016/j.enggeo.2018.04.013

**Reviewer #2:**  Dear Editor,

I appreciate the opportunity to review the manuscript titled "A multiscale comprehensive index for evaluating the development level of surface granite discontinuities," which was submitted to PLOS ONE. This manuscript aims to propose a index to estimate the develop level of the granite rock surface. The topics is of considerable interest to the scientific community and I hope that after the review, the authors will be able to resubmit the work. However, the logic of the manuscript is somewhat disorganized, and there are several language problems.

Here are some of my suggestions that I hope will help authors improve their manuscript.

1.I would like you to make a language edit. This is to improve the written English. It is mainly correct. However, there are strange phrases and technical terms used throughout the paper.

2.The authors are suggested to provide more information about the geological settings of the study area.

3.The quality of all the figures is required to be improved. For example, add scale and legends in fig.1.

4.The rock discontinuity is 3D geometry issue.“Introduction” must be literature review revised to be more in-depth and informative, with particular attention to other articles concerning similar approaches and other techniques used in literature with laser scanning; I permit to suggest some article, please see: https://doi.org/10.1007/s00603-021-02748-w;
https://doi.org/10.1007/s10706-020-01203-x

5.There are too many of figures, please remove the non-necessary figures or merge several figures to one figure, i.g., Figs. 5-7.

6.Please give more details about the field surveys in the manuscript, and it is a good way to show the photos for data collection in the field.

Best regards,

6. PLOS authors have the option to publish the peer review history of their article (what does this mean? ). If published, this will include your full peer review and any attached files.

**Do you want your identity to be public for this peer review?** For information about this choice, including consent withdrawal, please see our Privacy Policy .

Reviewer #1: No

Reviewer #2: No

---

## [Author Response · Author response to Decision Letter 1]

26 Mar 2025

Dear Reviewers:

Thank you for your letter and for the reviewers’ comments concerning our manuscript entitled “A multiscale comprehensive index for evaluating the development level of surface granite discontinuities” (ID: PONE-D-24-40516). Those comments are all valuable and very helpful for revising and improving our paper, as well as the important guiding significance to our researches. We have studied comments carefully and have made correction which we hope meet with approval. Revised portion are marked in red in the paper. The main corrections in the paper and the responds to the reviewer’s comments are as flowing:

Response to all reviewer’s comments:

Reviewer 1:

The manuscript presents a novel Multiscale Discontinuity Comprehensive (MDC) index for assessing the development level of surface granite discontinuities, which is crucial for the geological disposal of high-level radioactive waste. The study's focus on the Gansu Beishan area, a preselected candidate site for such a repository in China, adds regional significance. The topic is interesting, and this manuscript has a certain degree of innovation; however, some revisions are needed before it can be considered for publication. The detailed comments and suggestions are given as follows:

(1) Why is there a need to propose such a new metric? The explanation is not sufficiently clear.

Response: Thanks for the positive comment on the manuscript.

Regional and engineering-scale structural facets (faults) play a critical role in controlling the stability and continuity of engineering rock masses. Thus, a multiscale structural facet synthesis method is necessary to evaluate the degree of structural facet development in rock masses.

(2) Figure 9 is missing a legend.

Response: The legend has been added to Figure 9�Figure 9 has been modified to Figure 7 in the text�.

(3) Why is Eq. (5) used to express the new index in this form? Please further explain.

Response: Surface rock mass structural planes mainly consist of faults and joint. Faults are represented by the area of the fault zone. Joints are evaluated using the JSR development degree indicator. Therefore, we use the fault development indicator S and the joint development degree indicator JSR as the evaluation criteria for the development degree of surface rock mass structural planes.

(4) To be frank, the details and depth of analysis in the paper are insufficient. It is recommended to strengthen these aspects further.

Response: Thanks for the positive comment on the manuscript.

The revised manuscript has included relevant content and marked it in red. The Introduction has been expanded with reference questions. Geological characteristics of the study area have been added. A detailed description of the multi-scale structural surface comprehensive index MDC is provided.

(5) “Based on the characteristics of the fault mud observed on the wall of the exploratory trench (Figure 4), the fault's formation was deduced, and its spatial extension was traced and investigated using remote sensing maps and GPS to determine its development scale.” →“Based on the characteristics of the fault mud observed on the wall of the exploratory trench (Figure 4), the formation of the fault was deduced, and its spatial extension was traced and investigated using remote sensing maps and GPS to determine the scale of its development.”

Response: Thanks for the positive comment on the manuscript.

The revised manuscript has done the editing based on the question and is marked in red.

(6) “FIGURE10 Stress state of joint” → “FIGURE 10: Stress State of Joints”

Response: Thanks for the positive comment on the manuscript.

The revised manuscript has done the editing based on the question and is marked in red.

(7) Several key related papers should be considered for inclusion: (1) http://dx.doi.org/10.1016/j.enggeo.2016.03.007; and (2) https://doi.org/10.1016/j.enggeo. 2018.04.013

Response: Thanks for the positive comment on the manuscript. The work is very important for evaluating the development level of surface granite discontinuities. We have added the references in the context. The relevant papers have been cited.

Zheng, JunZhao, YuLu, QingDeng, JianhuiPan, XiaohuaLi, Yunzhen.A discussion on the adjustment parameters of the Slope Mass Rating (SMR) system for rock slopes[J].Engineering Geology, 2016, 206(Null).DOI:10.1016/j.enggeo.2016.03.007.

Zheng J , Yang X ,Qing Lü,et al.A new perspective for the directivity of Rock Quality Designation (RQD) and an anisotropy index of jointing degree for rock masses[J].Engineering Geology, 2018, 240.DOI:10.1016/j.enggeo.2018.04.013.

Reviewer 2:

I appreciate the opportunity to review the manuscript titled "A multiscale comprehensive index for evaluating the development level of surface granite discontinuities," which was submitted to PLOS ONE. This manuscript aims to propose a index to estimate the develop level of the granite rock surface. The topics is of considerable interest to the scientific community and I hope that after the review, the authors will be able to resubmit the work. However, the logic of the manuscript is somewhat disorganized, and there are several language problems.

Here are some of my suggestions that I hope will help authors improve their manuscript.

1. I would like you to make a language edit. This is to improve the written English. It is mainly correct. However, there are strange phrases and technical terms used throughout the paper.

Response: Thanks for the kind suggestion. We have carefully revised the manuscript according to the reviewers' comments, and also have asked a professional to review the English to check and improve the English.

2.The authors are suggested to provide more information about the geological settings of the study area.

Response: Thanks for the positive comment on the manuscript. Geological characteristics of the study area have been added in the revised manuscript and is marked in red.

3.The quality of all the figures is required to be improved. For example, add scale and legends in fig.1.

Response: Thanks for the positive comment on the manuscript. The figures has been redrawn and the scale and legend has been added.

4.The rock discontinuity is 3D geometry issue.“Introduction” must be literature review revised to be more in-depth and informative, with particular attention to other articles concerning similar approaches and other techniques used in literature with laser scanning; I permit to suggest some article, please see: https://doi.org/10.1007/s00603-021-02748-w;
https://doi.org/10.1007/s10706-020-01203-x.

Response: Thanks for the positive comment on the manuscript. The work is very important for evaluating the development level of surface granite discontinuities. We have added the references in the context. The relevant papers have been cited.

Ge, Y., Cao, B. & Tang, H. Rock Discontinuities Identification from 3D Point Clouds Using Artificial Neural Network. Rock Mech Rock Eng 55, 1705–1720 (2022). https://doi.org/10.1007/s00603-021-02748-w.

Farmakis I , Marinos V , Papathanassiou G ,et al.Automated 3D Jointed Rock Mass Structural Analysis and Characterization Using LiDAR Terrestrial Laser Scanner for Rockfall Susceptibility Assessment: Perissa Area Case (Santorini)[J].Geotechnical and Geological Engineering, 2020, 38(3):3007-3024.DOI:10.1007/s10706-020-01203-x.

5. There are too many of figures, please remove the non-necessary figures or merge several figures to one figure, i.g., Figs. 5-7.

Response: Thanks for the positive comment on the manuscript. The Figures 5, 6, and 7 have synthesizes a figure.

6. Please give more details about the field surveys in the manuscript, and it is a good way to show the photos for data collection in the field.

Response: Thanks for the positive comment on the manuscript. The more details about the field surveys has been added to the revised manuscript.

---

## [Decision Letter · Decision Letter 1]

8 Apr 2025

A multiscale comprehensive index for evaluating the development level of surface granite discontinuities

PONE-D-24-40516R1

Dear Dr. Liu,

We’re pleased to inform you that your manuscript has been judged scientifically suitable for publication and will be formally accepted for publication once it meets all outstanding technical requirements.

Kind regards,

Fabio Trippetta, Ph.D.

Academic Editor

PLOS ONE

Additional Editor Comments (optional):

Reviewers' comments:

Reviewer's Responses to Questions

**Comments to the Author**

1. If the authors have adequately addressed your comments raised in a previous round of review and you feel that this manuscript is now acceptable for publication, you may indicate that here to bypass the “Comments to the Author” section, enter your conflict of interest statement in the “Confidential to Editor” section, and submit your "Accept" recommendation.

Reviewer #1: All comments have been addressed

Reviewer #2: All comments have been addressed

2. Is the manuscript technically sound, and do the data support the conclusions?

Reviewer #1: Yes

Reviewer #2: Yes

3. Has the statistical analysis been performed appropriately and rigorously? 

Reviewer #1: Yes

Reviewer #2: Yes

4. Have the authors made all data underlying the findings in their manuscript fully available?

Reviewer #1: Yes

Reviewer #2: Yes

5. Is the manuscript presented in an intelligible fashion and written in standard English?

Reviewer #1: Yes

Reviewer #2: Yes

6. Review Comments to the Author

Reviewer #1: (No Response)

Reviewer #2: First of all, thanks editors very much to invite me again to review the revised version of the manuscript. After checking the revised manuscript, I would like to say that all of my concerns have been addressed by the authors, and the current version of manuscript can be considered acceptable.

7. PLOS authors have the option to publish the peer review history of their article (what does this mean? ). If published, this will include your full peer review and any attached files.

**Do you want your identity to be public for this peer review?** For information about this choice, including consent withdrawal, please see our Privacy Policy .

Reviewer #1: No

Reviewer #2: No

---

## [Editor Report · Acceptance letter]

PONE-D-24-40516R1

PLOS ONE

Dear Dr. Liu,

I'm pleased to inform you that your manuscript has been deemed suitable for publication in PLOS ONE. Congratulations! Your manuscript is now being handed over to our production team.

Kind regards,

on behalf of

Prof. Fabio Trippetta

Academic Editor

PLOS ONE